# Exposure to Ambient Air Pollution and the Incidence of Dementia in the Elderly of England: The ELSA Cohort

**DOI:** 10.3390/ijerph192315889

**Published:** 2022-11-29

**Authors:** Dylan Wood, Dimitris Evangelopoulos, Sean Beevers, Nutthida Kitwiroon, Klea Katsouyanni

**Affiliations:** 1Environmental Research Group, School of Public Health, Imperial College, London W12 0BZ, UK; 2MRC Centre for Environment and Health, Imperial College, London W12 0BZ, UK; 3NIHR HPRU in Environmental Exposures and Health, Imperial College, London W12 0BZ, UK; 4Medical School, National and Kapodistrian University of Athens, 11527 Athens, Greece

**Keywords:** dementia, air pollution, environmental risk factors

## Abstract

Increasing evidence suggests an adverse association between ambient air pollution and the incidence of dementia in adult populations, although results at present are mixed and further work is required. The present study investigated the relationships between NO_2_, PM_10_, PM_2.5_ and ozone on dementia incidence in a cohort of English residents, aged 50 years and older, followed up between 2004 and 2017 (English Longitudinal Study of Ageing; *n* = 8525). Cox proportional hazards models were applied to investigate the association between time to incident dementia and exposure to pollutants at baseline. Hazard ratios (HRs) were calculated per 10 μg/m^3^. Models were adjusted for age, gender, physical activity, smoking status and level of education (the latter as a sensitivity analysis). A total of 389 dementia cases were identified during follow-up. An increased risk of developing dementia was suggested with increasing exposure to PM_2.5_ (HR: 1.10; 95% confidence interval (CI): 0.88, 1.37), whilst NO_2_, PM_10_ and ozone exhibited no discernible relationships. Hazard ratios were 0.97 (CI: 0.89, 1.05) for NO_2_; 0.98 (CI: 0.89, 1.08) for PM_10_; 1.01 (CI: 0.94, 1.09) for ozone. In the London sub-sample (39 dementia cases), a 10 μg/m^3^ increase in PM_10_ was found to be associated with increased risk of dementia by 16%, although not statistically significant (HR: 1.16; CI: 0.90, 1.48), and the magnitude of effect for PM_2.5_ increased, whilst NO_2_ and ozone exhibited similar associations as observed in the England-wide study. Further work is required to fully elucidate the potentially adverse associations between air pollution exposure and dementia incidence.

## 1. Introduction

Dementia is a leading cause of death in England and Wales (12% of registered deaths [1]) and the World Health Organisation considers it a global public health priority [2]. Ageing populations and population growth suggest that the estimated 57.4 million sufferers worldwide in 2019 is projected to rise to 152.8 million by 2050 [3]. The estimated global cost associated with dementia also rose to USD 818 billion in 2015 and is predicted to increase alongside rising cases, with the steepest cost increases predicted for low- and middle-income countries [2].

A growing body of evidence suggests a number of environmental risk factors to be associated with declining cognition and increasing dementia incidence, including exposure to ambient air pollution [4]. However, a lack of understanding of the potential underlying biological mechanisms and pathways, as well as the inconsistent results reported so far [5,6], do not allow conclusive inference. Mechanistic evidence for adverse associations between NO_2_, particulate matter, ozone and dementia pathologies has been cited in previous studies, such as investigations into neuro-inflammation, oxidative stress [7], amyloid-β peptide aggregates and intracellular hyperphosphorylated tau protein aggregates in the brain [8] caused by increased exposure to ambient air pollution. Additionally, the incipient nature of dementia increases difficulty in accurately diagnosing any form of the condition, which may also prove problematic in detecting associations with environmental risk factors; given that accurately assigning a time frame for which clinical dementia develops to the point of diagnosis within an individual is of critical importance when attributing and quantifying their potential impacts on developing the condition [9,10].

Despite the historic lack of evidence pertaining to environmental risk factors and their association with cognitive decline and dementia incidence, particularly in comparison with other risk factors [11], there is increasing evidence to support a link between dementia incidence and exposure to ambient air pollution in adult populations [4,12]. A growing number of studies have reported adverse associations between exposure to several pollutants and the incidence of dementia, however results are mixed and considerable work is required in the coming years to fully understand the potential relationships. The present study utilised an English cohort of individuals aged 50 years and older, the English Longitudinal Study of Ageing (ELSA), in order to investigate the potential associations between exposure to ambient NO_2_, PM_10_, PM_2.5_, ozone and the incidence of dementia.

## 2. Materials and Methods

### 2.1. Study Population

ELSA is an ongoing cohort study aimed at assessing the dynamics of ageing in the elderly of England, and to inform debates in policy regarding health, wealth and lifestyles of the country’s elderly population utilising a representative sample [13]. The study was established in 2002 and is comprised of bi-annual waves following up those recruited at baseline, as well as those recruited through several refreshment samples. The ELSA study collects respondent data on daily activities, general health, cognition, chronic disease, wealth, smoking, alcohol intake, anthropometry and physical activity via in-person interviews, self-completion questionnaires and nurse visits. Respondents aged 50 years and older at baseline (‘core members’) from across England were included in the present study. The data made available included core members (*n* = 8525) from baseline through to the seventh follow-up wave of ELSA (2002–2017). A sub-sample of the main sample (*n* = 738), consisting of London-dwelling ELSA respondents only, was also analysed separately because residents of London have higher exposures at their residences on average.

### 2.2. Dementia Assignment

The ELSA study records self-reported doctor-diagnosed chronic conditions at each wave through interviews with either the respondent in question or a proxy interviewee. At each interview wave, the respondent or proxy is asked to observe a list of chronic conditions and state those for which the respondent has been diagnosed by a doctor. The present study utilised Wave 2 (2004–2005) of the ELSA study as baseline for dementia incidence investigation, including only those individuals that were reported as dementia-free. Wave 1 dementia data was not made available to the present study. Follow-up data up to Wave 8 (2016–2017) was made available. Respondents reporting to be diagnosed with either Alzheimer’s disease dementia or any other form of dementia (the two options provided during interviews) were both treated as an incident case in analysis (whether interviewed in-person or via proxy).

### 2.3. Modelled Concentrations of NO_2_, PM_10_, PM_2.5_ and Ozone at the Subjects’ Residence

The Community Multiscale Air Quality Urban (CMAQ-urban [14]) dispersion model incorporates the CMAQ [15] and Atmospheric Dispersion Modelling System Roads models to estimate hourly concentrations of several pollutants at a 20 × 20 m grid level. Residential postcode was the finest spatial resolution made available for linkage for each ELSA participant. CMAQ-urban estimates were, therefore, averaged for each pollutant at the postcode level by calculating the annual average concentration (from hourly modelled estimates) within the grid cell containing the postcode centroid. Annual average CMAQ-urban estimates of NO_2_, PM_10_, PM_2.5_ and ozone were assigned to ELSA participants at residential postcode level for the year 2004 as an estimate of baseline exposure. The fine resolution of CMAQ-urban has the ability to capture within-postcode variability, and estimates were modelled specifically for 2004 only, for the present study. Validation of CMAQ-urban 2004 modelled concentrations against measured concentrations for the same year is provided in Appendix A. Performance was assessed in both urban and rural areas across kerbside, roadside, rural, suburban and industrial monitoring sites for NO_2_ (*n* = 122 sites), PM_10_ (*n* = 88), PM_2.5_ (*n* = 10) and ozone (*n* = 69). Good performance in comparison with measured concentrations was observed for all pollutants, with *r* values ranging from 0.78 to 0.91 (Appendix A). Due to stipulations imposed by the managers of the ELSA data set (National Centre for Social Research; NatCen), pollutant estimates were classified into categories (deciles for NO_2_, PM_10_ and PM_2.5_; quintiles for ozone; Appendix A) in order to eliminate the possibility of ad-hoc identification of individuals’ postcodes through point estimate combinations. The mid-range value of each percentile range was then assigned to respondents as an estimate of baseline exposure.

### 2.4. Covariates

Covariate information included in analysis was decided upon through both a search of the relevant literature and the data made available to the present study. Variables cited as potential confounders or effect modifiers in previous studies of dementia incidence and air pollution exposure were included. Cox proportional hazards models were adjusted for baseline age, gender, smoking status and physical activity. Baseline smoking status was taken from interview data in which respondents identified themselves as currently smoking, being a former smoker or having never smoked. In terms of physical activity, respondents were asked on a scale of 1–5 how often they independently partake in vigorous, moderate and light physical activity in a typical week. The present study compiled each individuals’ responses and each category was weighted at a ratio of 3:2:1 (vigorous:moderate:light) to produce a composite variable in which respondents were then classified as “very active”, “moderately active” or “sedentary”. A total of 8525 ELSA respondents present in Wave 2 with complete covariate information were included in analysis. Sufficient markers of socioeconomic status were not made available to the present study for all respondents. As a sensitivity analysis, the age at which a respondent reported leaving full-time education was included in additional Cox proportional hazards models adjusted for all of the aforementioned covariates. Such education data was only available for participants within the sample that remained in the ELSA cohort until at least Wave 8 (*n* = 3920).

### 2.5. Statistical Analysis

Single pollutant Cox proportional hazards models were applied in order to investigate the relationship between exposure to each pollutant and dementia incidence. Each model independently assessed the associations between NO_2_, PM_10_, PM_2.5_ or ozone and was adjusted for the aforementioned covariates. All analysis was undertaken in R [16]. A dementia event was defined as the interview date in which a respondent (or proxy interviewee) stated that a doctor had diagnosed the respondent with dementia. Survival time was measured as time in years from baseline (rounded to the nearest six months) until end of follow-up in 2017, dropout, or a reported diagnosis of dementia (either by the respondent through an in-person interview or via a proxy interviewee). The same procedure was undertaken for both the England-wide sample and the London-only sub-sample.

## 3. Results

### 3.1. Demographic and Interview Data

Summary statistics for ELSA respondents included in analysis are provided in Table 1. The mean age at baseline was 64.4 years at Wave 2 and the sample was made up of 4703 women and 3822 men. Seven waves of the ELSA study from 2004 to 2017 were included in the linked data set and an average of 6.65 interviews were provided per individual. In terms of baseline smoking status, former and current smokers accounted for 63.5% of the total sample population. For physical activity, ELSA respondents classified as “moderately” or “very” active constituted 65.1% of the total sample population, with the remaining 34.9% reportedly “sedentary”. Of the 8525 ELSA individuals at baseline, approximately 4000 remained in the study until 2017.

### 3.2. Dementia Cases

For the purposes of the present study, a dementia case was defined as a self-reported doctor diagnosis of any form of dementia. A total of 389 ELSA respondents within the England-wide study sample reported having been diagnosed by a doctor with any form of dementia throughout the course of follow-up, accounting for 4.6% of the total number of respondents included in analysis. A mean follow-up time of 8.45 years was observed. In the London sample, 39 cases were observed (5.3%) with a mean follow-up time of 8.38 years.

### 3.3. Exposure to Ambient Air Pollution

Mean baseline concentrations for each pollutant assigned to ELSA respondents are provided in Table 2 for NO_2_ (25.5 μg/m^3^), PM_10_ (19.15 μg/m^3^), PM_2.5_ (12.13 μg/m^3^) and ozone (46.88 μg/m^3^). Correlation coefficients between pollutants are provided in Appendix A. No large differences were observed in exposure across respondent baseline age (Appendix A). Multi-pollutant models were conducted as a sensitivity analysis, for those pollutants with a Spearman correlation co-efficient <0.70, with no discernible change in hazards ratios observed (data not presented).

### 3.4. Dementia Incidence in Relation to Air pollution Exposure

Across single pollutant models, an increased risk of developing dementia was suggested for PM_2.5_ but did not reach the nominal level of statistical significance. No associations were identified for NO_2_, PM_10_ or ozone. The HR per 10 μg/m^3^ increase in pollutant concentrations were 0.97 (CI: 0.89, 1.05), 0.98 (CI: 0.89, 1.08), 1.10 (CI: 0.88, 1.37) and 1.01 (CI: 0.94, 1.09) for NO_2_, PM_10_, PM_2.5_ and ozone, respectively (Figure 1).

Analysis on the London sub-sample of ELSA participants within the present study (*n* = 738) showed no association for NO_2_ and ozone as in the England-wide study. An increased risk of dementia per 10 μg/m^3^ increase in PM_10_ was observed (HR = 1.16 [0.90, 1.48]), however the nominal level of significance was not reached. The magnitude of effect for PM_2.5_ increased in the London sub-sample (HR = 1.37 [0.95, 1.97]) compared with the England-wide study (*p* = 0.10; Appendix A).

In terms of adjusted covariate information in the models (Table 3), hazard ratios for each were consistent across independent single pollutant models. Being male was found to have an adverse association with developing dementia in the study sample, although 95% confidence intervals were wide and did not retain statistical significance. An increase of one year in baseline age was found to increase the risk of developing dementia by 14% across all models (CI: 1.12, 1.15). Self-reported levels of physical activity at baseline were also found to have a statistically significant relationship with dementia incidence, with those reportedly sedentary having an increased risk of developing dementia of 57% (CI: 1.26, 1.94) compared with those moderately active, whilst those reporting to be very active showing a reduced risk of 40% (CI: 0.40, 0.91; compared with those moderately active). Former smokers also observed a reduced risk of around 30% (CI: 0.51, 0.98; compared with current smokers). Protective associations were found across all models for the risk of dementia among never smokers compared with current smokers but did not reach the nominal level of statistical significance.

### 3.5. Sensitivity Analysis

The inclusion of education information as a further covariate to the models described in the main analysis was conducted on a sub-sample of the ELSA cohort data made available to the present study for which such information was viable (*n* = 3920; 143 dementia cases; Figure 2). The sub-sample comprised individuals that provided 1.18 interviews more on average, in comparison with the main sample, as well as individuals being younger by a mean of 3.75 years. Composition by gender and smoking status was similar between the two samples, with 5.6% more respondents reporting to be “very active” in the sub-sample. Mean baseline pollutant concentrations assigned to individuals were close to identical between the two samples. No hazard ratio (per 10 μg/m^3^ increase) changed by more than 13% across any of the single pollutant models in comparison with that of the main analysis (Figure 1), with no changes observed in direction of association or nominal statistical significance.

## 4. Discussion

The present study reports non-statistically significant associations between exposure to ambient air pollutants and the incidence of dementia in a sample population representative of the elderly of England, although for PM_2.5_ there was an indication that the risk for dementia was elevated with increasing exposure. These results remained after a sensitivity analysis on a sample of about 50% of participants with the inclusion of a marker for socioeconomic status (SES). In the London sub-sample of ELSA participants, the magnitude of effect for PM_2.5_ increased in comparison with the England-wide study (*p* = 0.10). The direction of association for PM_10_ was positive, however did not meet the nominal level of statistical significance. The inability to report a statistically significant relationship may suggest a lack of an association between the incidence of dementia in adult populations and exposure to ambient air pollution. However, the growing body of epidemiological evidence suggests adverse relationships between NO_2_, PM_10_, PM_2.5_, ozone and incident dementia to exist [4,12], although null findings and mixed results are present in the literature. Furthermore, the direction of associations reported in the present study are consistent with a number of previous investigations.

In studies investigating the relationship between NO_2_ exposure and dementia incidence, several investigations have reported increasing NO_2_ concentrations to be associated with an elevated risk of dementia incidence [1,17,18,19], whilst others have reported no associations [20,21] and one finding an adverse association [22]. Studies of PM_10_ exposure and dementia incidence have produced mixed results [18,20,22], with Li et al. (2019) finding an elevated risk of dementia incidence with increased PM_10_ concentrations [18].

In terms of PM_2.5_ exposure and dementia incidence, a number of studies have reported higher PM_2.5_ concentrations to be associated with an increased risk of dementia incidence [1,19,20,21,23,24,25,26]. However, Cerza et al. (2019) reported no association [22]. Shi et al. (2020) reported a 13% increase in dementia risk per 5 μg/m^3^ increase in PM_2.5_ concentrations in a U.S. cohort [26], whilst Jung et al. (2015) [20] reported a 38% elevation in risk per 4.4 μg/m^3^ increase in PM_2.5_ in Taiwan. The present study suggests a rise in PM_2.5_ concentrations to be associated with an elevated risk of developing dementia of 14% per 10 μg/m^3^ increase.

Previous work has suggested an association between increased ozone exposure and an elevated risk of developing dementia [20,22,23], whereas two studies failed to find any association [1,18]. Jung et al. (2015) reported an elevated risk of dementia incidence with increases in ozone concentrations in Taiwan [20], whilst Cerza et al. (2019) concluded that an increase in ozone concentration of 10 μg/m^3^ in Rome was associated with a 6% increase in risk of hospitalisation with dementia [22].

Most notably for the present study, Carey et al. (2018) found adverse relationships for NO_2_ (HR = 1.15 [1.04, 1.28] per 7.47 μg/m^3^ increase) and PM_2.5_ (HR = 1.06 [1.01, 1.13] per 0.95 μg/m^3^ increase) with dementia incidence and a protective association for ozone (HR = 0.85 [0.76, 0.96] per 5.56 μg/m^3^ increase) within a study sample of 130,978 individuals aged 50–79 at baseline registered across 75 London general practices between 2005 and 2013 [1]. The lack of formally significant findings in comparison with Carey et al. (2018) [1], despite similar baseline concentration estimates, may be due to the smaller sample size in the present study. However, the direction of associations for all pollutants in analysis of the London sub-sample in the present study are comparable to the findings of Carey et al. (2018).

The pollutants assessed in the present study are correlated, with each pollutant representing a different aspect of the pollution mixture. Particulate matter is more homogeneously distributed compared with NO_2_ and, thus, measured with less measurement error. Larger measurement error in NO_2_ assessment is linked to less statistical power to detect an increased risk [27] and may, thus, explain the lack of association. Additionally, CMAQ-urban uses the grid cell in which the postcode centroid is contained in order to calculate the annual average for a given postcode. Postcodes in the U.K. represent, on average, only 15 households. Their centroid point provides a good representation of population level exposure, as it is representative of where people live, rather than areas which might include large spaces without houses.

The procedure undertaken to assign residential postcode concentration estimates to ELSA respondent interview data required complete anonymity for cohort members. This resulted in the present study being precluded from undertaking post-hoc analyses regarding geographic identifiers at a higher resolution than government office region (GOR). Sensitivity analyses investigating the impact of living in urban areas in comparison with rural areas may have provided further context for the London sub-study, however, the inability to perform such analyses must, therefore, be noted as a potential limitation.

The lack of relevant SES markers is also a limitation of the present study. However, the results of the sensitivity analysis inclusive of data on age at which left education, a marker of SES which is often used, suggests that education markers did not alter findings of the principal analyses. The possibility of selection bias cannot be excluded because subjects with information on education had longer follow-up and were younger compared with the whole sample. Furthermore, several other potential confounders were not made available to the present study which may have provided stronger evidence for associations. Nutritional patterns, high body mass index, excessive alcohol intake, medication use and genetic factors such as the presence of the APOE-ε4 allele have all been cited as risk factors for dementia which will need to be included in future epidemiological work and the omission of such information here is a limitation of the present study.

A potential advantage of the present study is the inclusion of cohort respondent data spanning up to 15 years. Long follow-up periods should provide greater statistical power given the ability to monitor events over the course of study. However, the potential for measurement error in pollutant concentration estimates may be increased given the idea that exposure to pollutant concentrations may be more likely to change in longer term studies.

As posited in previous work, the potential for loss to follow-up in cohort studies may be increased in respondents that develop dementia during follow-up [28]. Such a pattern may be likely given the debilitating nature of accelerated cognitive decline and its potentially prohibitive impact in continuing to be an active member of an ongoing cohort. Thus, subjects with longer follow-up possibly have a lower risk of developing dementia and further work is required in order to ascertain the potential introduction of bias via this process of selective dropout. However, the bias in the effect estimates is likely towards null as cases are lost if selective drop out is occurring.

Dementia assignment in ELSA may also highlight a potential limitation of the present study. ELSA respondents are asked to identify having ever been diagnosed with Alzheimer’s disease dementia or any other form of dementia. Positive responses to either of these two possibilities are coded separately, with other forms of dementia such as Lewy body dementia, which accounts for 5–10% of dementia cases, undecipherable from the interview data. The ability to analyse vascular dementia specifically would have been of particular interest to the present study due to the broad range of previous work linking cardiovascular health outcomes to air pollution exposure, as well as studies, such as Li et al. (2019), linking increased NO_2_ and ozone exposure to an elevated risk of developing vascular dementia specifically [18]. The use of self-reported health conditions in cohort studies has the potential to introduce classification error, particularly with conditions affecting memory such as dementia. Additionally, utilising ELSA recorded data as a proxy for time of dementia diagnosis creates further complications and uncertainties in assigning survival times and censorship. Accurately diagnosing the onset of dementia is inherently difficult due to its incipient nature and reliance upon such individuals then informing interviewers within a bi-annual longitudinal cohort study provides further complications, particularly in assigning appropriate exposure windows.

The use of the Informant Questionnaire on Cognitive Decline in the Elderly (IQCODE) score based on proxy responses to assign dementia cases was not conducted in the present study. This method has been used in previous studies (e.g., [29,30]), however, the requisite data was not available to the present study. Previous studies of dementia incidence in ELSA exhibit a wide range of reported dementia cases within the cohort, even amongst those investigating similar follow-up periods (Appendix A). The reasons behind such disparities are unclear and suggest that accurate dementia assignment in ELSA respondents is difficult and may highlight a limitation of the present study, potentially leading to an underestimation of dementia cases within the cohort and non-significant results.

The strong associations reported for age and physical activity across all models are plausible and in line with previous work. Sedentary behaviour was strongly linked to an increased risk of developing dementia, with higher levels of activity reportedly beneficial. Age is the biggest risk factor for dementia onset and such findings are not surprising in a cohort of respondents (aged 50 years and older at the time of recruitment) followed up for up to 15 years in the present study.

## 5. Conclusions

The present study provides suggestive evidence for an association between increasing exposure to PM_2.5_ and an elevated risk in the incidence of dementia, however no associations were reported for NO_2_, PM_10_ or ozone. These results may provide further evidence for an emerging field of epidemiological research, particularly in a relatively understudied region. The inability to report any statistically significant association may be plausibly attributed to a lack of accurate dementia diagnosis data, decreased accuracy in exposure estimates due to the process required to convert point estimates into ordinal variables or the current lack of understanding of the relevant exposure windows [12]. Existing work has produced mixed results to date, and the results presented here suggest an association and point to the need for further investigation of this important public health issue.

## Figures and Tables

**Figure 1 ijerph-19-15889-f001:**
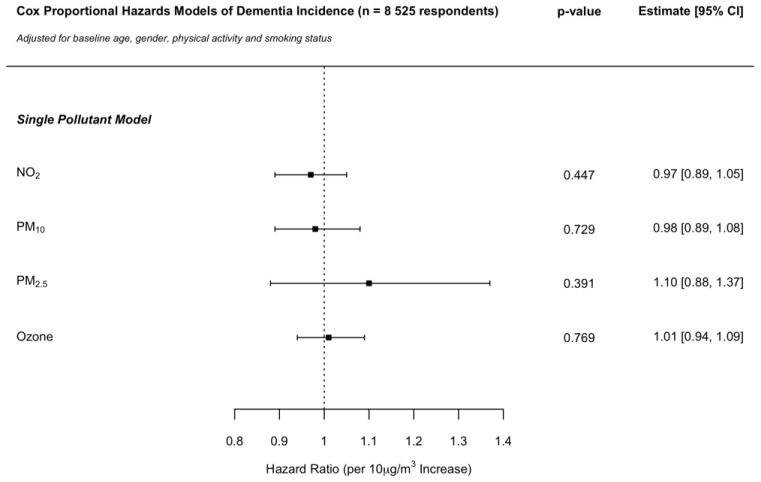
Cox proportional hazards model hazard ratios per 10 μg/m^3^ increase in NO_2_, PM_10_, PM_2.5_ and ozone concentrations on dementia incidence in the ELSA cohort across England. Independent single pollutant models adjusted for baseline age, gender, physical activity and smoking status.

**Figure 2 ijerph-19-15889-f002:**
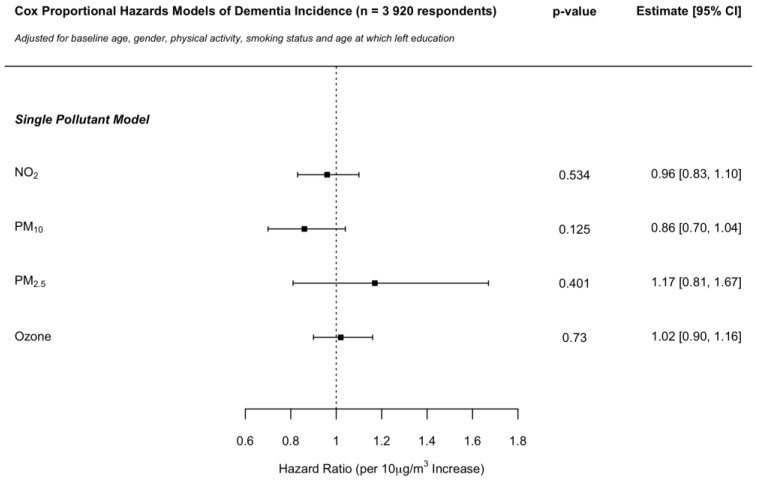
Sensitivity analysis: Cox proportional hazards model hazard ratios per 10 μg/m^3^ increase in NO_2_, PM_10_, PM_2.5_ and ozone concentrations on dementia incidence in the ELSA cohort across England. Independent single pollutant models adjusted for baseline age, gender, physical activity, smoking status, and age at which left education.

**Table 1 ijerph-19-15889-t001:** Baseline (Wave 2) descriptive statistics for demographic, interview and covariate data for ELSA respondents included in analysis of dementia incidence.

Demographic, Interview and Covariate Data	ELSA Participants
**Interview Data**	
Number of respondents	8525
Mean number of interviews provided per participant (±SD)	6.65 (±1.87)
**Demographic information**	
* Gender*	
Number of women (% of respondents)	4703 (55.2%)
Number of men (% of respondents)	3822 (44.8%)
* Baseline Age (Years)*	
Mean (±SD)	64.4 (±9.76)
Range	50–93
**Baseline Covariate Information**	
* Physical Activity*	
Sedentary (% of respondents)	2972 (34.9%)
Moderately active (% of respondents)	4193 (49.2%)
Very active (% of respondents)	1360 (15.9%)
* Smoking Status*	
Never smoked (% of respondents)	3108 (36.5%)
Former smoker (% of respondents)	3952 (46.4%)
Current smoker (% of respondents)	1465 (17.1%)
* Age Left Full-Time Education (n = 3920)*	
14 or under/Never went to school (% of respondents)	433 (11%)
At 15 (% of respondents)	1364 (34.9%)
At 16 (% of respondents)	808 (20.6%)
At 17 (% of respondents)	329 (8.4%)
At 18 (% of respondents)	244 (6.2%)
19 or over (% of respondents)	659 (16.8%)
Not yet finished (% of respondents)	83 (2.1%)

**Table 2 ijerph-19-15889-t002:** Baseline CMAQ-urban modelled pollutant concentrations linked to ELSA respondents included in analysis of dementia incidence.

Pollutant	Mean Baseline CMAQ-Urban Concentration (μg/m^3^) ±SD
NO_2_	25.5 ± 13.7
PM_10_	19.15 ± 10.51
PM_2.5_	12.13 ± 4.41
Ozone	46.88 ± 13.53

**Table 3 ijerph-19-15889-t003:** Cox proportional hazards ratios (confidence intervals) for covariates included in dementia incidence analysis of ELSA respondents across England.

Covariate	Hazard Ratio (95% CI)
**Gender**	
Female	Reference category
Male	1.04 (0.84, 1.29)
**Baseline Age (per year increase)**	1.14 (1.12, 1.15)
**Baseline Physical Activity**	
Sedentary	1.57 (1.26, 1.94)
Moderately active	Reference category
Very active	0.60 (0.40, 0.91)
**Baseline Smoking Status**	
Former smoker	0.71 (0.51, 0.98)
Current Smoker	Reference category
Never smoked	0.84 (0.60, 1.16)
**Age Left Full-Time Education (*n*** = **3920)**	
14 or under/never went to school	0.87 (0.54, 1.41)
At 15	Reference category
At 16	0.83 (0.51, 1.36)
At 17	1.35 (0.73, 2.47)
At 18	0.52 (0.19, 1.45)
19 or over	0.69 (0.38, 1.26)
Not yet finished	0.61 (0.54, 1.41)

## Data Availability

The English Longitudinal Study of Ageing (ELSA) was developed by a team of researchers based at University College London, the Institute for Fiscal Studies and the National Centre for Social Research. The data are linked to the UK Data Archive and freely available through the UK data services and can be accessed at: https://beta.ukdataservice.ac.uk/datacatalogue/studies/study?id=5050 (accessed on 10 November 2017).

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
