# Peer review of "Exposure to Ambient Air Pollution and the Incidence of Dementia in the Elderly of England: The ELSA Cohort"

_ijerph, 2022, doi:10.3390/ijerph192315889_

Round 1
Reviewer 1 Report (Previous Reviewer 1)
Generally, the authors have addressed reviewer comments well.
However, I have a few additional (minor) comments:
1. The authors assigned the air pollution estimates from the CMAQ models by annual average concentration of the grid cell containing the postcode centroid for a given postcode, however, each zip code area should consist of several grid cells (20m*20m) from the CMAQ model. It is unclear why the authors did not choose to use the average of the annual average concentrations of all grid cells that were located in the postal code area rather than just the postal code with the centroid. Using the centroid may have increased the measurement error for the pollutants that are less homogeneously distributed due to local sources. This might also be a reason why only PM2.5 which is relatively more homogenous than NO2 and PM10 is the only pollutant suggesting increased risks. This issues should also be mentioned when they discuss the lack of formally statistically significant results.
2. The air pollution measures (PM 2.5, PM10, NO2, O3) estimated are not independent measures but rather markers for complex mixtures and sources that are correlated. Thus, it would be helpful to state this much more clearly and produce a more comprehensive view of air pollution effects on the brain especially given the measurement error issues that differ between locally heterogenous and more homogeneously distributed pollutants (see #1).
3. I don’t understand why the first sentence in the discussion was added “The present study reports non-statistically significant associations between zone, NO2, PM10, PM2.5 and ozone on the incidence of dementia in a sample population representative of the elderly of England” as the only suggested association is seen for PM2.5, while for the other pollutants there were no associations observed or even suggested. This is also acknowledged in the first sentence of the conclusions that reads ” In this study we observed no statistically significant associations between exposure to pollutants and The relationships observed between exposure to each pollutant and the incidence of dementia in England, although an indication of increased risk with increasing exposure to PM2.5 was seen.” Please be consistent! Similarly, this sentence makes no sense lines 288-89 “However, the direction of associations for all comparable pollutants may strengthen the findings of Carey et al. (2018)….”
4. Please specify why in line 318 you state that “Further work is required in order to ascertain the potential introduction of bias via selective dropout, however the bias in the effect estimates is likely towards the null as cases are lost if selective drop out is occurring.” i.e. in what manner selective?
5. Please reword line 216 “Insignificant findings “ what do you mean by insignificant…
Author Response
Please note: the line numbers on my version seem to differ from those provided by the reviewers, therefore I have referenced modifications and edits relative to sections and paragraph numbers to avoid confusion.
- The authors assigned the air pollution estimates from the CMAQ models by annual average concentration of the grid cell containing the postcode centroid for a given postcode, however, each zip code area should consist of several grid cells (20m*20m) from the CMAQ model. It is unclear why the authors did not choose to use the average of the annual average concentrations of all grid cells that were located in the postal code area rather than just the postal code with the centroid. Using the centroid may have increased the measurement error for the pollutants that are less homogeneously distributed due to local sources. This might also be a reason why only PM2.5 which is relatively more homogenous than NO2 and PM10 is the only pollutant suggesting increased risks. This issues should also be mentioned when they discuss the lack of formally statistically significant results.
Response: The reasons for using postcode centroids are explained here: Postcodes in the UK do not relate to areas (comparable to, for example, zip code areas in the U.S.) but represent number of houses (delivery points) which share the same postcode for mail delivery purposes (on average 15 households per postcode). The postcode centroid represents the mid-point of the delivery points which share the same postcode. For example, in a street of 15 houses with the same postcode, the postcode centroid would be the house in the middle. The postcode centroid therefore provides a good representation of population level exposure, as it is representative of where people live, rather than areas without houses or parks.
In the new paragraph 6 of the discussion information has been included in order to convey this to the reader. The text states:
The pollutants assessed in the present study are correlated, with each pollutant representing a different aspect of the pollution mixture. Particulate matter is more homogeneously distributed compared to NO2 and thus measured with less measurement error. Larger measurement error in NO2 assessment is linked to less statistical power to detect an increased risk [27] and may thus explain the lack of association.
- The air pollution measures (PM 2.5, PM10, NO2, O3) estimated are not independent measures but rather markers for complex mixtures and sources that are correlated. Thus, it would be helpful to state this much more clearly and produce a more comprehensive view of air pollution effects on the brain especially given the measurement error issues that differ between locally heterogenous and more homogeneously distributed pollutants (see #1).
Response: Paragraph 6 of the discussion was added to address this comment. The text states:
The pollutants assessed in the present study are correlated, with each pollutant representing a different aspect of the pollution mixture. Particulate matter is more homogeneously distributed compared to NO2 and thus measured with less measurement error. Larger measurement error in NO2 assessment is linked to less statistical power to detect an increased risk [27] and may thus explain the lack of association.
- I don’t understand why the first sentence in the discussion was added “The present study reports non-statistically significant associations between zone, NO2, PM10, PM2.5 and ozone on the incidence of dementia in a sample population representative of the elderly of England” as the only suggested association is seen for PM2.5, while for the other pollutants there were no associations observed or even suggested. This is also acknowledged in the first sentence of the conclusions that reads ” In this study we observed no statistically significant associations between exposure to pollutants and The relationships observed between exposure to each pollutant and the incidence of dementia in England, although an indication of increased risk with increasing exposure to PM2.5 was seen.” Please be consistent! Similarly, this sentence makes no sense lines 288-89 “However, the direction of associations for all comparable pollutants may strengthen the findings of Carey et al. (2018)….”
Response: Thank you for this comment and apologies for the inconsistencies. Reporting of the results throughout has been modified to be more consistent. The following modifications have been made:
The first sentence of the discussion has been modified to the following:
The present study reports non-statistically significant associations between exposure to ambient air pollutants and the incidence of dementia in a sample population representative of the elderly of England, although for PM2.5 there was an indication that the risk for dementia was increased with increasing exposure.
Paragraph 5 of the discussion has been modified to the following:
However, the direction of associations for all pollutants in analysis of the London sub-sample in the present study are comparable to the findings of Carey et al. (2018).
The first sentence of the conclusions has been modified to the following:
The present study provides suggestive evidence for an association between increasing exposure to PM2.5 and an elevated risk in the incidence of dementia, however no associations are found for NO2, PM10 or ozone
- Please specify why in line 318 you state that “Further work is required in order to ascertain the potential introduction of bias via selective dropout, however the bias in the effect estimates is likely towards the null as cases are lost if selective drop out is occurring.” i.e. in what manner selective?
Response: Thank you for this comment. The following sentence has been added to address this in paragraph 10 of the discussion:
Thus, subjects with longer follow-up possibly have a lower risk of developing dementia and further work is required in order to ascertain the potential introduction of bias via this process of selective dropout.
- Please reword line 216 “Insignificant findings “ what do you mean by insignificant…
Response: Thank you for your comment, this has now been modified to the following:
Protective associations were found across all models for the risk of dementia among never smokers compared to current smokers but did not reach the nominal level of statistical significance
Reviewer 2 Report (New Reviewer)
Wood et al. reported on the results of a population-based cohort study (ELSA) aimed at determining the association between exposure to ambient NO2, PM, and O3 and the incidence of dementia among a representative sample of elderly individuals from England followed-up from 2002 to 2017.
Methods
The study design (a prospective cohort) is an appropriate approach to evaluate the hypothesis at hand. The exposure was modelled (CMAQ and Atmospheric Dispersion Modelling System Roads) on an hourly basis and at 20x20 m spatial resolution, and then annually averaged and spatially aggregated at postcode level for 2004 (defined as baseline). Notably, estimates of exposure showed a "good performance" in comparison to measured concentrations at monitoring sites. By design, exposed and unexposed cohorts, more exactly, the gradient of exposure across individuals, can be considered representative of the target population.
Moreover, individuals (or their proxies) who reported a previous doctor-diagnosis of dementia were excluded from the risk population and the follow-up time (mean: 8.5 years) was long enough to observe incident cases. From lines 153-154 can be inferred that losses to follow-up were small, however, this report would benefit from a statement regarding absolute and relative frequencies of participants who dropped out the study, and a contrast of those against the total cohort to inform the possibility (and degree) of the risk of selection bias. Also, considering the self-reported nature of the outcome, it would be helpful to the readers to have a sense of the magnitude and direction of information bias in relation to, for example, a diagnosis based on the audit of clinical records from other studies: In the presence of non-directional misclassification of the outcome, underestimation of the magnitude of associations is expected (to the null); however, in scenarios of directional misclassification the authors should elaborate/speculate about their potential impact on the findings.
In terms of confounding, the authors adjusted for relevant covariates; however, the lack of socioeconomic status (SES) information might have biased (underestimated) estimates of association due to residual confounding. Although a sensitivity analysis was conducted by using education attainment as a proxy for SES, in pursuing a reduction in the risk of confounding, it could have accentuated selection bias whenever the subsample with education data differs from the original cohort: I suggest adding information that allow the readers to determine whether the sensitivity analysis contributed or not to the minimizing both selection bias and confounding. Finally, how was the proportionality of hazards tested? This, considering the election of Cox regression (a semi-parametric modelling approach) as the analytic tool.
Results
This section clearly shows the main findings of the study. No further comments on it.
Discussion/Conclusion
This section summarizes the main findings of the study and contrasts them against the available, relevant evidence. Strengths and limitations are also stated; however, I suggest including a more elaborated discussion about limitations derived from losses to follow-up and a sensitivity analysis based on education as proxy of SES.
Author Response
Please note: the line numbers on my version seem to differ from those provided by the reviewers, therefore I have referenced modifications and edits relative to sections and paragraph numbers to avoid confusion.
Note: Reviewer 2 provides a summary of the paper, but has embedded suggestions for modification to which we have responded.
Wood et al. reported on the results of a population-based cohort study (ELSA) aimed at determining the association between exposure to ambient NO2, PM, and O3 and the incidence of dementia among a representative sample of elderly individuals from England followed-up from 2002 to 2017.
Methods
The study design (a prospective cohort) is an appropriate approach to evaluate the hypothesis at hand. The exposure was modelled (CMAQ and Atmospheric Dispersion Modelling System Roads) on an hourly basis and at 20x20 m spatial resolution, and then annually averaged and spatially aggregated at postcode level for 2004 (defined as baseline). Notably, estimates of exposure showed a "good performance" in comparison to measured concentrations at monitoring sites. By design, exposed and unexposed cohorts, more exactly, the gradient of exposure across individuals, can be considered representative of the target population.
Moreover, individuals (or their proxies) who reported a previous doctor-diagnosis of dementia were excluded from the risk population and the follow-up time (mean: 8.5 years) was long enough to observe incident cases. From lines 153-154 can be inferred that losses to follow-up were small, however, this report would benefit from a statement regarding absolute and relative frequencies of participants who dropped out the study, and a contrast of those against the total cohort to inform the possibility (and degree) of the risk of selection bias. Also, considering the self-reported nature of the outcome, it would be helpful to the readers to have a sense of the magnitude and direction of information bias in relation to, for example, a diagnosis based on the audit of clinical records from other studies: In the presence of non-directional misclassification of the outcome, underestimation of the magnitude of associations is expected (to the null); however, in scenarios of directional misclassification the authors should elaborate/speculate about their potential impact on the findings.
In terms of confounding, the authors adjusted for relevant covariates; however, the lack of socioeconomic status (SES) information might have biased (underestimated) estimates of association due to residual confounding. Although a sensitivity analysis was conducted by using education attainment as a proxy for SES, in pursuing a reduction in the risk of confounding, it could have accentuated selection bias whenever the subsample with education data differs from the original cohort: I suggest adding information that allow the readers to determine whether the sensitivity analysis contributed or not to the minimizing both selection bias and confounding. Finally, how was the proportionality of hazards tested? This, considering the election of Cox regression (a semi-parametric modelling approach) as the analytic tool.
Response: Information was added on the completeness of follow-up in the final line of Section 3.1:
Of the 8,525 ELSA individuals at baseline, approximately 4,000 remained in the study until 2017.
Additionally, the follow text was added to Section 3.5:
The sub-sample comprised of individuals that provided 1.18 interviews more on average, in comparison to the main sample, as well as individuals being younger by a mean of 3.75 years. Composition by gender and smoking status was similar between the two samples, with 5.6% more respondents reporting to be “very active” in the sub-sample. Mean baseline pollutant concentrations assigned to individuals were close to identical between the two samples.
And in paragraph 8 of the discussion:
The possibility of selection bias cannot be excluded because subjects with information on education had longer follow-up and were younger compared to the whole sample.
Results
This section clearly shows the main findings of the study. No further comments on it.
Discussion/Conclusion
This section summarizes the main findings of the study and contrasts them against the available, relevant evidence. Strengths and limitations are also stated; however, I suggest including a more elaborated discussion about limitations derived from losses to follow-up and a sensitivity analysis based on education as proxy of SES.
Thank you for your comments. The following modifications have been made to address them:
Paragraph 10 of the discussion:
Thus, subjects with longer follow-up possibly have a lower risk of developing dementia and further work is required in order to ascertain the potential introduction of bias via this process of selective dropout.
Please also see previous responses above.
This manuscript is a resubmission of an earlier submission. The following is a list of the peer review reports and author responses from that submission.
Round 1
Author Response
I like to ask the authors to temper their sentiment and interpret the data in a more comprehensive and statistical significance driven manner – as well as acknowledge the limitations as the most important criteria in interpreting their study results. I.e. the results need to be presented in the context of the study limitations from the outset and less credence should be given to statistical significance, as, for example, the effect estimates presented would contribute a positive association for PM2.5 to a meta-analysis.
This comment is much appreciated and has been addressed throughout the revised manuscript, in the abstract (lines 20-23), results (lines 192-196), discussion (lines 280-288) and conclusion.
What is currently missing the most is a discussion of the exposure modeling validity given that exposure was estimated for different size postal code areas across the UK and for only one year.
The model is finely resolved at 20x20m and therefore has the ability to capture the variability in concentrations at postcode level. The model was run for 2004 only and not across multiple years, specifically for the present study. Section 2.3 now contains a discussion on this (Lines 93-101).
The potential impact of selective loss to follow-up of individuals with dementia who might also have lived in more deprived neighborhoods and, thus, been exposed more highly; i.e. it is likely that the most exposed dementia-affected individuals did not return to self-report their dementia diagnosis (a well - known phenomenon in dementia studies in general i.e. that the cognitive impairment contributes to loss to follow-up). This is especially concerning as the authors did not have access to proxy respondent reports either that may make up to some degree for this issue. I suggest to further investigate the issues of informative loss to follow-up (who was lost etc) and that this issue is being discussed more extensively in terms of its potential for the introduction of bias.
Proxy interview data was included in the present study (please see the last sentence of Section 2.2). We completely agree that loss to follow-up may likely be higher in those with dementia and this has now been highlighted in the discussion (Lines 312-325).
Study population: The authors used data from the UK based ELSA cohort. The average age of the participants at baseline was 64 and the cases average follow-up years was 8 years. Is there any more information available whether study subjects lived in urban vs. rural areas in addition to how many individuals were from London? Clearly, the effect estimates for PM2.5 and PM10 were much stronger (albeit not formally statistically significant) for the 738 individuals from London. The sources for particulate matter in urban areas are likely different from rural areas, thus it would be interesting to combine urban area residents in another possibly more informative (due to larger sample size than the London only analysis) sensitivity analysis.
As stated above, the anonymous linkage process precluded any post-hoc analyses regarding geographic information. The highest resolution geographic identifier linked to ELSA respondents after linkage of exposure data was Government Office Region (GOR). England is made up of nine GORs which encompass both urban and rural areas and therefore an urban vs rural sensitivity analysis was not possible. London is a GOR itself and thus this analysis was possible. This has now been added to the discussion section (Lines 289-295).
Dementia diagnosis: The ELSA respondents were asked to identify having ever been diagnosed with Alzheimer’s disease dementia or any other form of dementia at each biennial follow-up visit. The authors did not have access to proxy respondent reports though. Thus, there is a good chance for selection bias due to informative loss to follow-up of the most exposed diseased. Please address this issue and provide some data for those lost to follow-up. Additionally, please describe what date was used to define an “event” for the purpose of the Cox model. Were events the date of visit? The midpoint between visits? Was the time-scale collapsed to increments of 1 calendar year?
Proxy interview data was included in the present study. Events were defined as the interview date in which a respondent stated that they been diagnosed with dementia. The time to event was rounded to the nearest six months from baseline. We realise that this was not made clear enough in the text and this has been amended in Section 2.5 (Lines 134-136).
The respondents’ residential address-based exposure was derived from the air pollution concentrations estimated from a CMAQ models at a 20*20m spatial resolution. However, it seems that the authors used the postal(zip?) code only and not the full address to assign exposure? If so, each zip code area should consist of several grid cells from the CMAQ model. How did the authors treat the exposure concentrations for each grid cell and which value did the authors assign to each respondent? Please provide more details on the exposure assignment for each respondent. Additionally, Line 88: “The mid-range value of each band was then assigned to respondents as an estimate of baseline exposure”. Please clarify what “band” means, is this the percentile range?
Residential postcode was the finest spatial scale of residential information made available to the present study for linkage. The process involved calculating the annual average concentration of the grid cell containing the postcode centroid for a given postcode (for all ~2.1 million postcodes in England). These estimates were given to the ELSA data managers who then linked the ELSA respondents anonymously and removed postcode information before sending the linked data set back to the authors. The process through which exposure data was linked has been updated in the text in Section 2.3 (Lines 98-102). Apologies that the word “band” was used, we agree that this is a confusing and incorrect term. It is the percentile range and this has been addressed (Line 106).
For a more than 10-year period follow-up, the residence-based exposure might reflect the true exposure level less and less correctly, especially amongst the younger enrollees, as they would be expected to be exposed at the workplace i.e. do effect estimates differ by age group at enrolment possibly due to larger exposure misclassification in younger enrollees? Did the authors take into account the mobility of the respondents?
We agree that this is a limitation. The benefits of having such a long follow-up period in terms of the number of potential events and statistical power are positive, however this does also present the limitation that long-term estimates may not necessarily reflect exposure accurately over the course of follow-up and may induce measurement error. This is however a common approach in epidemiological studies of potentially adverse effects of air pollution exposure on health. Baseline exposure did not differ greatly between respondents when categorised into 10-year age categories Please see Table S2 which has now been included in the supplementary material. We were unable to take mobility into account given the lack of such data available. This has now been addressed as a limitation in the discussion (Lines 306-311).
Please present and discuss the correlation pattern between the air pollutants. The authors calculated the effect estimates for each air pollutant and dementia in single-exposure models only. Why did the authors not repeat the analyses using co-exposure models to address the potential confounding by other air pollutants? And did you consider the confounding by meteorological factors such as temperature for O3?
Please see the correlation matrix below. This has now been referenced in text and placed in the Supplementary Material Table S3 (Lines 160-165). Multi-pollutant models were run for instances in which the Spearman correlation coefficient was less than 0.7.
Please discuss the validity and possible difference in estimating exposures with the CMAQ model for urban and rural areas and by postal code area size and what the consequences may be; possibly conduct a sensitivity analysis by postal code area size (small areas may have much more accurate exposure assignments than larger areas).
As previously stated, this was not possible in the present study given the anonymous exposure data linkage process. This has been highlighted in text in the discussion section (Lines 293-295).
Table 3 presentation of covariate associations with the outcome is confusing as one would expect all associations to be more or the same – i.e. why not present the overall associations for covariates without pollutants in the model. The way the estimates are currently presented by pollutant model may mislead some readers to think of them as a stratified model i.e. that the estimates represent pollutant estimates and not covariate estimates. In fact, some sensitivity analyses by age at baseline (<70, >=70) might be warranted and should be presented.
Following this comment, Table 3 has now been edited to include covariate hazard ratios and confidence CIs (overall associations from a model run without pollutants as suggested). As highlighted above, very little difference was observed in exposure estimates across baseline age groups and so this sensitivity analysis was not conducted.
Throughout the manuscript, please do NOT refer to ‘effects’, but rather ‘associations’ or ‘effect estimates’ instead.
Thank you for this comment. The manuscript now describes ‘associations’ rather than ‘effects’ in all relevant sections.
Reviewer 2 Report
There are some publications that have been studied and found association between ambient air pollution and dementia although results at present are mixed and further work is required. I give a few revisions to authors to possibly improve the quality of the paper.
1. Air pollutant and dementia are both important issues around the world. The hypothesis or possible mechanisms should be introduced between air pollutant and the incidence of dementia. However, it is unclear why air pollutants exposure might increase/ associate with the dementia in this article.
2. Dementia assignment was based on self-reported doctor-diagnosed chronic conditions by interviewing in-person or via proxy. It might cause serious recall bias and classification bias which need to be stated clearly in the limitation.
3. The main exposure air pollutants should be illustrated and quantified. Exposure measurement would be the most important issue to be understood. However, it is unclear to get those concentration definition, for example, daily average, month average, 8-hours max ozone average and so on.
4. Exposure of air pollution was measured by residential postcode at baseline year. However, this group of elderly might have high probability of moving house between the study period (2004-2017), maybe you should check /calculate the exposure of air pollution base on their residential postcode in every wave.
5. What was the rational of classifying air pollutants into different categories (deciles for NO2, PM10 and PM2.5; quintiles for ozone)?
6. What was the characteristic of ELSA cohort, e.g., method of recruitment, sampling, or interview, which suggest you state a brief description in your article?
7. In the study, you chose the second wave (2004-2005) as your major study group. What was the difference between Wave 1 and Wave 2?
8. Potential confounders of dementia such as age, gender, smoking status and physical activity had been controlled in your model, however, there was still other important confounders needed to be considered in your study like medical and medication history, genetic factor, or social interaction. In the other hand, Nutrition/ Obesity factor would be an important issue in this study.
Author Response
- Air pollutant and dementia are both important issues around the world. The hypothesis or possible mechanisms should be introduced between air pollutant and the incidence of dementia. However, it is unclear why air pollutants exposure might increase/ associate with the dementia in this article
Thank you for this comment. The introduction section now includes a brief overview of posited biological pathways through which ambient air pollution may increase the risk of dementia (Lines 41-46).
- Dementia assignment was based on self-reported doctor-diagnosed chronic conditions by interviewing in-person or via proxy. It might cause serious recall bias and classification bias which need to be stated clearly in the limitation.
This has been addressed in the discussion and cited as a potential limitation (Lines 319-324).
- The main exposure air pollutants should be illustrated and quantified. Exposure measurement would be the most important issue to be understood. However, it is unclear to get those concentration definition, for example, daily average, month average, 8-hours max ozone average and so on.
A better definition has been provided in Section 2.3 (lines 98-102) to inform the reader that long term estimates were calculated, i.e., annual average concentrations for each pollutant calculated from hourly modelled concentrations.
- Exposure of air pollution was measured by residential postcode at baseline year. However, this group of elderly might have high probability of moving house between the study period (2004-2017), maybe you should check /calculate the exposure of air pollution base on their residential postcode in every wave.
The incipient nature of dementia, coupled with a lack of a current understanding on the relative exposure windows, led us to investigate long term exposure to air pollution in relation to dementia incidence and the baseline 2004 concentrations can be regarded as a good proxy for that. This is in contrast to short term exposure changes between ELSA waves (every two years). Additionally, due to the anonymous data linkage process, we did not have the opportunity after linkage to know which individuals moved and which didn’t.
- What was the rational of classifying air pollutants into different categories (deciles for NO2, PM10 and PM2.5; quintiles for ozone)?
Estimates were categorised in order to abide by the ELSA data managers’ stipulations that no individual may be identified post-linkage with the exposure data set. This required categorising estimates so that none of the 2.1 million postcodes in England modelled for the study were unique or near-unique in terms of concentration estimate combinations (which would allow post-hoc identification of individuals’ postcodes). Ozone provided less heterogeneity in point estimates and was suitable to be categorised into quintiles. This allowed for 5 less categories and therefore a lower likelihood of unique combinations. This trade-off allowed for the inclusion of more individuals in the study (see lines 101-107)
- What was the characteristic of ELSA cohort, e.g., method of recruitment, sampling, or interview, which suggest you state a brief description in your article?
Following this comment we amended Section 2.1. The characteristics of interest are provided in Table 1. In terms of sampling and recruitment, we have given a brief overview in Section 2.1 and cited the Wave 1 Technical Report as a reference. This allowed us to provide these answers whilst also remaining as concise as possible in the text.
- In the study, you chose the second wave (2004-2005) as your major study group. What was the difference between Wave 1 and Wave 2
Dementia diagnosis data for Wave 1 was not provided and this is discussed in Section 2.2.
- Potential confounders of dementia such as age, gender, smoking status and physical activity had been controlled in your model, however, there was still other important confounders needed to be considered in your study like medical and medication history, genetic factor, or social interaction. In the other hand, Nutrition/ Obesity factor would be an important issue in this study
The present study utilised all relevant potential confounders where the data allowed it. Data on APOE-4 allele, nutrition, social interaction etc. was not made available. Thank you for highlighting this issue and it has been made clearer as a limitation in the discussion section (Lines 299-305).